# ONLINE VARIANCE-REDUCING OPTIMIZATION

**Nicolas Le Roux**
Google Brain
nlr@google.com

**Reza Babanezhad**
University of British Columbia
rezababa@cs.ubc.ca

**Pierre-Antoine Manzagol**
Google Brain
manzagop@google.com

## ABSTRACT

We emphasize the importance of variance reduction in stochastic methods and propose a probabilistic interpretation as a way to store information about past gradients. The resulting algorithm is very similar to the momentum method, with the difference that the weight over past gradients depends on the distance moved in parameter space rather than the number of steps.

## 1 INTRODUCTION

We are interested in minimizing $\ell(\theta) = E_x[\ell(\theta, x)]$, which we shall denote the *true error*. To that extent, we have access to a stream of examples $x_1, \ldots, x_t, \ldots$, coming in sequential order. At any time step $t$, we shall define the training error to be

$$\bar{\ell}_t(\theta) = \frac{1}{t} \sum_i \ell(\theta, x_i) . \tag{1}$$

Most optimization algorithms aim at accelerating the convergence speed of this minimization problem by improving its conditioning. This can be done by building a low-rank approximation of the inverse Hessian, like in L-BFGS (Nocedal & Wright, 2006), by casting the Newton direction as the solution of a quadratic problem approximately solved (Martens, 2010), by finding the steepest direction in an appropriate Riemannian space (Amari, 1998) or approximation thereof (Martens & Grosse, 2015), or by building online diagonal approximation of the Hessian (LeCun et al., 1998; Duchi et al., 2011; Kingma & Ba, 2014).

Another line of research looked at ways to reduce the variance of stochastic updates in order to boost convergence speeds. Indeed, the convergence rate of the original stochastic gradient algorithm (Robbins & Monro, 1951) depends on that variance (see, e.g., Theorem 6.2 from Bubeck et al. (2015)). In particular, a lot of work has been done in the finite training set setting to obtain stochastic methods with linear convergence rate for strongly convex problems, e.g. SAG (Le Roux et al., 2012; Schmidt et al., 2017), SDCA (Zhang et al., 2013; Shalev-Shwartz & Zhang, 2013), SVRG (Johnson & Zhang, 2013), SAGA (Defazio et al., 2014) and MISO (Mairal, 2013).

Interestingly, there has been little work in extending this approach to the online setting, most of it involving a growing batch strategy (Babanezhad et al., 2015; Hofmann et al., 2015), with the exception of TONGA (Le Roux & Fitzgibbon, 2010) which built an online estimate of the covariance matrix. However, the algorithm was too brittle to be of practical use.

Another class of algorithms aimed at robustifying stochastic gradient uses an average of past gradients to perform each update. This is for instance the case of momentum (Goh, 2017) which applies the following update to $\theta$:

$$\theta_t = \theta_{t-1} - \gamma_t \sum_{i=1}^{t} \alpha^{t-i} g(\theta_i, x_i) . \tag{2}$$

that is the update is a weighted combination of all past gradients where the weights are exponentially decreasing. While this decreases the variance, it does so only by a constant amount, and thus the variance of the updates does not converge to 0, requiring $\gamma_t$ to decrease to 0 to ensure convergence.

Momentum has often been presented as a way to fight curvature (see, e.g., (Goh, 2017)). However, in an online setting, while $g(\theta_t, x_t)$ is an unbiased estimate of $g(\theta_t)$, the variance of this estimate can be

arbitrarily large and does not go down to 0. A potential strategy is to gather more samples to reduce the variance of the update while keeping its unbiasedness, as proposed for instance by Friedlander & Schmidt (2012) in the finite setting. We will now show how a specific instance of momentum can indeed be seen as doing variance reduction.

## 2 HANDLING STOCHASTICITY

The variance of gradients at $\theta$ is $C(\theta) = E_x[\|g(\theta, x) - g(\theta)\|^2]$. While this bears resemblance with the Fisher information matrix, the expectation is taken under the data distribution, not the model distribution. It is sometimes called the **empirical Fisher matrix** (Martens, 2014) but contains very different information. For instance, in the case of the quadratic function $\ell(\theta, x_i) = \|\theta - x_i\|^2/2$, the Hessian and the Fisher information matrix are both equal to the identity matrix $I$ while the covariance $C$ is equal to the covariance of the $x_i$'s. Thus, one can easily be changed without changing the others.

Now that we have shown that the covariance matrix need not be related to the Hessian nor to the Fisher matrix, it remains to be seen how to use it to do variance reduction. In particular, we would like to find an algorithm whose updates have a variance going to 0, which would make convergence possible with a fixed stepsize, eliminating one of the main hassles of online methods.

Our approach will be centered around the reuse of old gradients. As these gradients have been computed for other values than $\theta_t$, the resulting update will be biased and one must trade off variance for bias. This is the strategy used in SAG (Le Roux et al., 2012), where the variance of the updates goes down to 0 despite the batch size being constant [1].

Early on in the optimization, $\theta$ will change quickly and old gradients will greatly increase the bias. Further, the variance of the gradients tends to be smaller compared to the mean in the early iterations, reducing the interest of batching examples. Close to convergence, however, a different situation occurs. Not only is the variance much larger than the average gradient, $\theta$ moves much less and the incurred bias of using old gradients is small.

Thus, an intuitive strategy would be to keep few gradients early on then to progressively increase the effective batch size as optimization progresses. Since we would like the variance to decrease as $1/t$, this means the effective batch size should grow linearly with $t$.

### 2.1 TRACKING THE TRUE GRADIENT

Every time we sample a function $\ell(\cdot, x)$ and compute its gradient in $\theta$, we get more information about the gradient of the average function $\bar{\ell}$. The idea is to carry over this information from sample to sample to get an increasingly accurate estimate of the true gradient.

Let us start with a non-informative prior: $g(\theta_0) \sim \mathcal{N}(0, +\infty)$. We then compute $g(\theta_0, x_1)$. Assuming the gradients are distributed according to a Gaussian of constant variance $C$, we get

$$g(\theta_0)|g(\theta_0, x_1) \sim \mathcal{N}(g(\theta_0, x_1), C) . \tag{3}$$

We then perform an update $\delta_1$ to get $\theta_1 = \theta_0 + \delta_1$. If we have an upper bound $L$ on the Hessian of $\ell$, we can approximate the new posterior by

$$g(\theta_1)|g(\theta_0, x_1) \sim \mathcal{N}\left(g(\theta_0, x_1) + \frac{L+\mu}{2}\delta_1, C + \frac{(L-\mu)^2}{4}\|\delta_1\|^2 I\right) . \tag{4}$$

The posterior in Eq. (4) differs from the posterior in Eq. (3) in two ways:

- The mean has been shifted to account for the displacement $\delta_1$ in parameter space. Ideally, the mean would be shifted by $H\delta_1$ but $H$ is unknown so an approximation based on the upper and lower bounds is used instead. Shifting the mean is critical for convergence.

- The variance of the posterior increases by $\frac{(L-\mu)^2}{4}\|\delta_1\|^2$ to account for the uncertainty in the Hessian. We see that this increase in variance is driven both by the difference between $\mu$ and $L$, which is directly related to the uncertainty around the true Hessian, and by the length of the displacement $\|\delta_1\|$ which compounds the uncertainty in the Hessian.

---

[1] SAGA (Defazio et al., 2014) is more careful and reduces the variance while keeping the updates unbiased.

We may then sample $x_2$, compute $g(\theta_1, x_2)$ and update our posterior. Assuming $\mu = 0$ to unclutter notation, we get $g(\theta_1)|\{g(\theta_0, x_1), g(\theta_1, x_2)\} \sim \mathcal{N}(\mu_1, C_1)$ with

$$\mu_1 = \left(2C + \frac{L^2}{4}\|\delta_1\|^2 I\right)^{-1} \left(C\left(g(\theta_0, x_1) + \frac{L}{2}\delta_1\right) + \left(C + \frac{L^2}{4}\|\delta_1\|^2 I\right)g(\theta_1, x_2)\right) \quad (5)$$

$$C_1 = \left(2C + \frac{L^2}{4}\|\delta_1\|^2 I\right)^{-1}\left(C^2 + \frac{L^2}{4}\|\delta_1\|^2 C\right) . \quad (6)$$

Analyzing the mean and the covariance of that posterior in more detail, we observe that:

- When we do a large parameter update, $\|\delta_1\|$ is very large and we get $\mu_1 \approx g(\theta_1, x_2)$ and $C_1 \approx C$. The posterior is centered around the new gradient, which makes sense since, because of the large move in parameter space and uncertainty around the Hessian, the old gradient does not provide any information anymore. This dynamic will be observed early on in the optimization where large progress is made and our method will be equivalent to pure stochastic gradient.

- With a small parameter update, $\|\delta_1\| << 1$ and $\mu_1 \approx \frac{1}{2}(g(\theta_0, x_1) + g(\theta_1, x_2))$ and $C_1 \approx \frac{C}{2}$. The posterior is the average of the two computed gradients since, due to the very small parameter update, the old gradient has low bias but helps decrease the variance. We see that the variance is halved compared to using just the last computed gradient. This is the regime in which we will be as we get closer to a local optimum and the variance of our posterior will converge to 0 at a rate $1/t$.

Thus, this algorithm automatically transitions from pure stochastic gradient to gradient averaging and it does so at a speed which depends on the covariance of the gradients and the size of the move in parameter space. This is similar to acceleration which uses an increasing momentum of the form $1 - 3/t$ but where the increase depends on the specificity of the problem rather than being predetermined.

## 3 EXPERIMENTS

We only have preliminary experiments on both a linear regression and a logistic regression problem. Fig. 1 shows that our methods seems to converge for a constant stepsize when both stochastic gradient and momentum plateau. We see that the first results seem to indicate a convergence, even

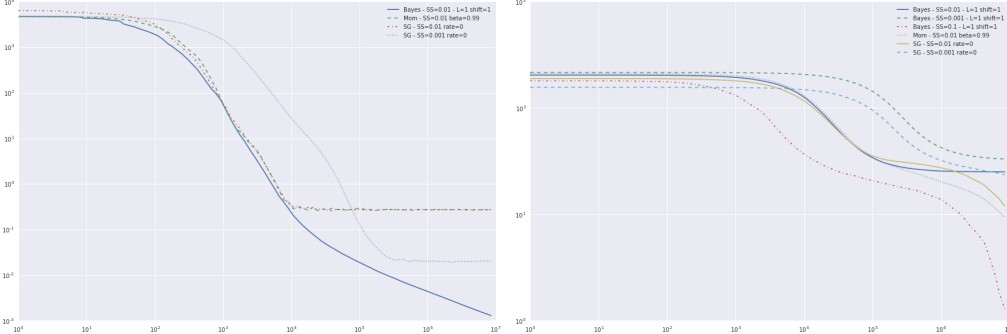

Figure 1: Comparison of stochastic gradient, momentum and Bayes optimizer on a linear regression (left) and a logistic regression (right) problems.

for a fixed stepsize. However, we assumed here knowledge of $L$ and $C$. It is yet unclear how this algorithm would perform when these quantities have to be estimated. Finally, a convergence proof might help find appropriate values for the constant stepsize.

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
