# OpenReview forum: "Online variance-reducing optimization"
_ICLR.cc/2018/Workshop — Accept_

### Official Review · AnonReviewer3 · 2018-03-06
**interesting interpretation of stochastic gradient methods**

**Rating:** 6
**Confidence:** 2

**Review:**

The paper provides a probabilistic interpretation of stochastic gradient methods that use past gradients. The algorithm is similar to moment methods but differs in weighting past gradients.

pros:
1) probabilistic interpretation is interesting.
2) preliminary experimental results look promising.

cons:
1) pseudo code of the algorithm and derivation of equations, which will make the paper easier to understand, are missing.
2) exposition is not very clear.

Some questions and concerns:
1) is \delta_1 sampled from (3) ?
2) does it need to store all the past gradients?
3) linearly growing batch size seems not good.
4) it may be nontrivial to get L, mu and C.

---

### Official Review · AnonReviewer2 · 2018-03-10
**Interesting idea for online SGD to allow constant stepsizes**

**Rating:** 8
**Confidence:** 3

**Review:**

This workshop note discusses adapting momentum (for SGD) in a specific way to, as they write, "automatically transition from pure stochastic gradient to gradient averaging" and it takes the variance of the gradient estimate to zero, allowing one to use a constant stepsize.

The idea seems quite promising, and I think this is a perfect workshop paper. The idea seems recent, and not fully fleshed out yet. Details and comparisons to other online methods need to be done for a longer version of the paper.

The paper itself could use some rewriting to make it clear. The actual algorithm is not written out explicitly; the figure is basically illegible; and the main objects of interest, g(theta,x) and g(x), are not even defined.  That said, the high level idea is clear, and presumably the details would be clearly presented in a later journal paper.

---

### Decision · Program_Chairs · 2018-03-20
**ICLR 2018 Workshop Acceptance Decision**

**Decision:**

Accept

**Comment:**

Congratulations, your paper was accepted to the ICLR workshop.